

# Longitudinal changes in estimated glomerular filtration rates in chronic hepatitis B patients treated with Tenofovir Alafenamide *vs* Entecavir

Xuan Li[1,2,*], Qiang Wu[3,*], Fang Huang[4], Changxiang Lai[1], Fengjuan Chen[5], Juan Meng[3], Fang Wang[1], Hui Zeng[4,6] and Lina Zhang[1,4]

[1] Department of Liver Diseases, Shenzhen Third People's Hospital, Shenzhen, Guangdong, China
[2] Phase I Clinical Trial Centre, Shenzhen Third People's Hospital, Shenzhen, Guangdong, China
[3] Department of Geriatric Medicine, Shenzhen Third People's Hospital, Shenzhen, Guangdong, China
[4] Medical Affairs Department, Shenzhen Third People's Hospital, Shenzhen, Guangdong, China
[5] Department of Infectious Diseases, Shenzhen Third People's Hospital, Shenzhen, Guangdong, China
[6] Health Management Department, Shenzhen Third People's Hospital, Shenzhen, Guangdong, China
[*] These authors contributed equally to this work.

Corresponding authors
Hui Zeng, 24178743@qq.com
Lina Zhang, chillsnow@163.com

## ABSTRACT

**Background.** For individuals with chronic hepatitis B (CHB) at higher risk of nephrotoxicity, entecavir (ETV) and tenofovir alafenamide (TAF) are recommended antiviral options. This study aimed to investigate kidney safety among treatment-naïve individuals with CHB receiving TAF versus ETV.

**Method.** Treatment-naïve individuals with CHB receiving either TAF or ETV from July 2019 to December 2020 were included. Follow-up data on estimated glomerular filtration rates (eGFR) were collected. Factors related to chronic kidney disease (CKD) development were analyzed by Cox regression analysis. Generalized additive mixed model (GAMM) was employed to investigate temporal eGFR changes and the association between the extent of follow-up eGFR change and antiviral agents.

**Results.** 466 treatment-naïve individuals with CHB were included, with 296 in the ETV group and 170 in the TAF group. In the subgroup of individuals with a baseline eGFR higher than 90 mL/min/1.73 m$^2$, 13.9% in the ETV group developed CKD, compared to 9.8% in the TAF group ($p = 0.304$). Multivariable Cox analysis demonstrated that male (hazard ratio (HR) 2.72; 95% confidence interval (CI) [1.02–7.25]; $p = 0.045$) and baseline eGFR (HR 0.86; 95% CI [0.82–0.90]; $p < 0.001$) were significantly associate with the CKD development. GAMM revealed that eGFR initially decreased and then stabilized around week 40. Every 12 weeks, the TAF group exhibited an overall lower rate of eGFR decline compared to the ETV group, with an adjusted difference of 0.38 mL/min/1.73 m$^2$ (95% CI [0.11–0.65], $p = 0.006$). The difference remained significant in males and patients over 35 years old.

**Conclusion.** The kidney safety profile of TAF among treatment-naïve individuals with CHB was comparable to that of ETV, without significant difference in developing CKD. Stratified analyses further revealed that TAF demonstrated superior kidney benefits compared to ETV specifically in males or patients aged over 35 years.

## INTRODUTION

Approximately 296 million people worldwide are infected by chronic hepatitis B (CHB), and 1.5 million new infections occur annually (*World Health Organization, 2023*). Beyond its higher prevalence, CHB is intricately linked to the occurrence of complications, including cirrhosis (*Perz Joseph et al., 2006*) and hepatocellular carcinoma (*El-Serag Hashem, 2012*), resulting in over 500,000 deaths annually. As a result, antiviral therapy has become the primary treatment of CHB, intending to suppress virus replication and induce regression of fibrosis (*European Association for the Study of the Liver, 2017*; *Korean Association for the Study of the Liver (KASL), 2022*).

International guidelines recommend first-line antiviral agents, including tenofovir alafenamide (TAF), tenofovir disoproxil fumarate (TDF), or entecavir (ETV), for CHB treatment owing to their potent virus inhibition and high resistance barrier (*European Association for the Study of the Liver, 2017*; *Terrault et al., 2018*; *Sarin et al., 2016*). Considering that complete eradication of HBV is rare, a majority of patients need life-long antiviral therapy, raising apprehensions regarding potential drug-related adverse effects. Prolonged administration of TDF has been linked to kidney toxicity, evidenced by a marked reduction in estimated glomerular filtration rate (eGFR) (*Tsai et al., 2018*; *Mak et al., 2022*; *Jung et al., 2022b*). Consequently, current guidelines advocate for the preference of TAF or ETV for individuals with an elevated risk of kidney impairment, including those with kidney alterations such as eGFR <60 mL/min/1.73 m$^2$ or undergoing hemodialysis (*European Association for the Study of the Liver, 2017*; *Korean Association for the Study of the Liver (KASL), 2022*; *Terrault et al., 2018*; *Drafting Committee for Hepatitis Management Guidelines & the Japan Society of Hepatology, 2020*; *You et al., 2023*).

Studies have documented that ETV and TAF are less likely to cause kidney function deterioration compared to TDF (*Pilkington et al., 2020*; *Han et al., 2017*). However, the introduction of TAF in the Chinese market in December 2018 has led to limited availability of clinical real-world research data. Besides, there is a lack of direct comparisons of the kidney outcomes in individuals with CHB between TAF and ETV, with conflicting conclusions in the existing literature.

Therefore, we carried out this clinical study aiming to investigate kidney outcomes among treatment-naïve individuals with CHB who received TAF or ETV, as well as to determine potential disparities across distinct demographic populations.

## METHOD

### Study design

This retrospective cohort study included treatment-naïve individuals with CHB receiving TAF or ETV from July 1st, 2019 to December 31st, 2020 at Shenzhen Third People's

Hospital in Guangdong China. CHB diagnosis, treatment status, and relevant clinical and laboratory data were obtained through medical chart reviews.

## Study participants

Treatment-naïve CHB individuals (18 years or older) receiving monotherapy with ETV or TAF for over 48 weeks were included if they had at least one follow-up creatine measurement per 48 weeks until August 31st, 2022. This follow-up frequency reflects the variability in clinical follow-up intervals in real-world settings and aligns with the KDIGO 2012 Clinical Practice Guideline for the evaluation and management of chronic kidney disease (CKD) (*Kidney Disease: Improving Global Outcomes, KDIGO, 2013*), which recommends at least one annual eGFR assessment for CKD monitoring. Additionally, the Chinese guidelines (*You et al., 2023*) for CHB management suggest that kidney function be tested every 6–12 months.

The criteria for exclusion included: (1) under 18 years old; (2) pregnancy at baseline or during follow-up; (3) decompensated liver disease or liver transplantation; (4) alcoholic liver disease (*Varga et al., 2017*); (5) co-infection of other hepatitis viruses like hepatitis C and E virus; (6) presence of chronic kidney diseases (defined as: eGFR under 60 mL/min/1.73 m$^2$, persistent $\geq$ 3 months before baseline (*Kidney Disease: Improving Global Outcomes, KDIGO, 2013*), or documented clinical diagnosis with International Statistical Classification of Diseases and Related Health Problems, 10th Revision codes of N00 to N29); patients with baseline eGFR <60 mL/min/1.73 m$^2$ with unknown persistent time are typically referred to nephrologists and receive specific therapy. To minimize confounding due to baseline kidney impairment and treatment, individuals with baseline eGFR <60 mL/min/1.73 m$^2$ were excluded; (7) co-infection of other pathogens undergoing treatment, such as human immunodeficiency virus, mycobacterium and parasites; (8) malignancy; (9) severe diseases of other systems: organ transplantation, acute and chronic failure of vital organs, acute events necessitating emergent intervention (*e.g.*, acute ischemic stroke, acute abdomen); (10) death during follow-up; (11) unknown baseline eGFR; (12) missing data on follow-up creatinine: cases with no creatinine measurements during any 48-week interval; (13) irregular treatment of ETV or TAF (Fig. 1).

The institutional review board at Shenzhen Third People's Hospital approved this research study (Ethical Reference number: 2022-142-02, 2022-142-03). In view of the retrospective design, informed consent was not required and was therefore waived by the institutional review board.

## Data collection

Demographics, medical prescriptions, and laboratory information at the initiation of ETV or TAF treatment were extracted from electronic health records, and considered as baseline measurements. All subsequent measurements post-treatment initiation were considered as follow-up data.

Accompanying disease diagnosis: (1) diabetes: fasting serum glucose level of at least 6.1 mmol/L (*Society Chinese Diabetes, 2021*) or current use of hypoglycemic agents. (2) Hypertension: systolic blood pressure $\geq$ 140 mmHg, diastolic blood pressure $\geq$ 90 mmHg, or

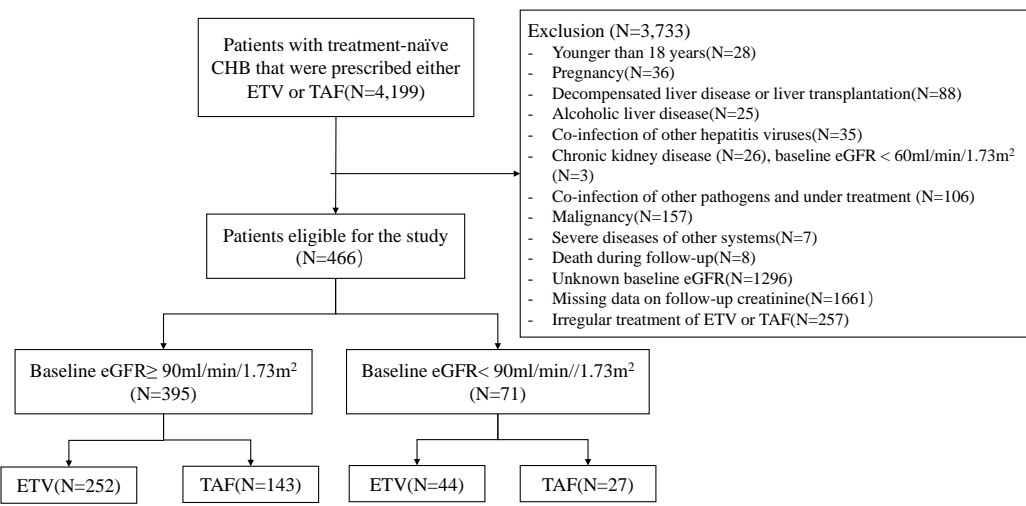

**Figure 1** Flow diagram of the study.

current usage of antihypertensive medications. (3) Dyslipidemia: low-density lipoprotein cholesterol level of at least 3.4 mmol/L, total cholesterol level of at least 5.2 mmol/L, triglyceride level of at least 1.7 mmol/L (*Joint committee issued Chinese guideline for the management of dyslipidemia in adults, 2018*), or current use of hypolipidemic agents. (4) Metabolic dysfunction-associated steatotic liver disease (MASLD) was determined by various reliable methods, including imaging techniques (magnetic resonance imaging, computed tomography, and ultrasonography), histological evidence, controlled attenuation parameter, and/or serum diagnostic markers (hepatic steatosis index or fatty liver index).

The diagnosis of cirrhosis was diagnosed through a comprehensive assessment that includes histological evidence, clinical history, imaging findings, endoscopic evaluations, laboratory findings, and signs of portal hypertension. These indicators may encompass a nodular appearance on imaging, splenomegaly, thrombocytopenia (platelets count lower than $100 \times 10^9$/L), hypoalbuminemia(serum albumin (ALB) level lower than 35 g/L), a prolonged prothrombin time of more than 3 s over control, presence of varices, and hepatic decompensation symptoms such as ascites, variceal hemorrhage and hepatic encephalopathy (*You et al., 2023*).

## Kidney outcomes

Kidney outcomes were evaluated using the Chronic Kidney Disease Epidemiology Collaboration (CKD-EPI) equation (*Levey et al., 2009*) and categorized according to the criteria specified by the Kidney Disease: Improving Global Outcomes (KDIGO) (*Kidney Disease: Improving Global Outcomes, KDIGO, 2013*). CKD progression was characterized by a sustained increase in the CKD stage lasting for a minimum of three consecutive months.

## Statistical analysis

Given that the proportion of missing data was less than 10%, missing data were imputed with median values (Fig. S1).

Categorical variables were presented as numbers (%) and compared by Chi-squared test or Fisher exact test. Continuous variables were expressed as means (standard deviations) and compared by t test if they followed a normal distribution; otherwise, they were presented as medians (interquartile ranges) and analyzed by Kruskal-Wallis test.

Factors influencing CKD development over time were explored using univariable Cox analysis. Collinearity among variables was tested using the variance inflation factor (VIF), and any variables with high collinearity (VIF > 10) were excluded from the Cox regression analysis. Creatinine (Cr) and total bilirubin (TB) demonstrated high VIF and were consequently excluded from the Cox analysis. Multivariable Cox analysis was conducted on variables that showed statistical significance ($p < 0.5$) in the univariable Cox analysis. Proportional hazard assumptions were validated by conducting the proportional hazard test based on Schoenfeld residual using the Cox.zph function in the R package of *survival*.

To investigate temporal changes in eGFR and the association between change rates in follow-up eGFR and treatment groups, generalized additive mixed model (GAMM) was employed, considering repeated measurements of follow-up eGFR and irregular measurement intervals (*Lin & Zhang, 1999*). The model fitted nonlinear smooth functions of time for overall and treatment groups, using cubic regression splines, to capture nonlinear eGFR trajectories. To assess potential effect modification by sex and age, interaction terms between treatment groups (TAF *vs.* ETV) and sex (male *vs.* female) or age group (≥35 *vs.* <35 years) were included in the GAMM models.

To quantify temporal changes in eGFR for specific intervals (0–12, 12–24, 24–48, and 48–86 weeks. 86 was the median follow-up duration), average slopes of eGFR (mL/min/1.73 m²/week) were calculated for the treatment groups, using the fitted smooth functions. Then GAMM with smooth curve fitting was used to assess the overall differences in eGFR change between treatment groups, both with and without covariates. Sex- and age-stratified analyses were conducted to investigate variations in eGFR between the two treatments across diverse demographic subgroups.

Sensitivity analyses were conducted to evaluate the stability of the primary results as follows: (1) a piecewise linear mixed model (LMM) was fitted to confirm the 12-week decline rate. This model included fixed effects for group and time (0–12 weeks, 12–24 weeks, 24–48 weeks, 48–68 weeks), as well as random intercepts and slopes for each subject. (2) Multivariable Cox analysis was conducted in the subgroup of eGFR ≥ 90 mL/min/1.73 m² without missing baseline data. (3) GAMM analyses were conducted on the following groups of individuals: those without any missing baseline data; those with eGFR measurements within 12 and 30 weeks, respectively; and those who had at least three eGFR measurements during the periods of 0 to 12 weeks, 12 to 24 weeks, and 24 to 48 weeks.

Statistical significance was determined as $p$ value < 0.05. The analyses were carried out in R software (version 4.1.3).

## RESULTS

### Baseline characteristics

According to the screening flow (Fig. 1), 466 treatment-naïve individuals with CHB were included with 296 receiving ETV and 170 receiving TAF for statistical analysis (Table 1).

At baseline, the ETV group demonstrated a higher median age (39.5 *vs.* 36.5, $p = 0.019$) and a greater proportion of male participants (79.1% *vs.* 67.6%, $p = 0.009$) compared to the TAF group. Both groups had statistically similar proportions of comorbidities, including hypertension, diabetes mellitus, dyslipidemia, non-alcoholic fatty liver disease (NAFLD), and compensated cirrhosis (all $p > 0.05$). Medication usage for hypertension, dyslipidemia, diuretics, and propranolol did not vary between the two groups ($p > 0.05$), although the ETV group had a higher percentage of hypoglycemic agent usage (6.4% *vs.* 1.8%, $p = 0.040$).

Moreover, when compared to the TAF group, the ETV group demonstrated elevated median levels of aspartate aminotransferase (AST) (45.0 *vs.* 41.0 U/L), TB (15.4 *vs.* 14.1 μmol/L), direct bilirubin (DB) (4.8 *vs.* 4.3 μmol/L), and gamma-glutamyl transferase (GGT) (41.0 *vs.* 34.0 U/L) (all $p < 0.05$). Conversely, there were no differences in ALT, ALB, Cr, and baseline eGFR (all $p > 0.05$). The proportion of detectable HBV DNA was similar between the two agents ($p = 0.068$), while a higher percentage of positive HBeAg was noted in the TAF group (61.8% *vs.* 48.6%, $p = 0.008$).

395 patients were in the subgroup of baseline eGFR $\geq$ 90 mL/min/1.73 m$^2$, and 71 were in the subgroup of baseline eGFR between 60 and 90 mL/min/1.73 m$^2$ (Table 1). The characteristics of the former subgroup were largely comparable to those of the overall group, with the exception of a non-significant difference in median age ($p = 0.074$) and the proportion of hypoglycemic drugs ($p = 0.171$). Within the subgroup of individuals with a baseline eGFR between 60 and 90 mL/min/1.73 m$^2$, the average age was significantly lower in the TAF group (44.4 *vs.* 51.3, $p = 0.015$), while other variables were found to be statistically similar.

### Kidney outcomes during follow-up

The follow-up durations were similar between the TAF and ETV groups in both the entire cohort and subgroups (all $p > 0.05$) (Table 1). Specifically, in the subgroup of baseline eGFR $\geq$ 90 mL/min/1.73 m$^2$, 14 (9.8%) patients in the TAF group and 35 (13.9%) in the ETV group developed CKD stage 2, with no statistically significant difference observed ($p = 0.304$). Furthermore, no patients in either treatment group experienced progression beyond CKD stage 2. Univariable Cox analyses revealed that age, sex, AST, ALB, and baseline eGFR were related to CKD development during the follow-up. However, only the male sex (hazard ratio (HR) 2.72; 95% confidence interval (CI) [1.02–7.25]; $p = 0.045$) and baseline eGFR (HR 0.86; 95% CI [0.82–0.90]; $p < 0.001$) were found to be independently associated with CKD development in multivariable Cox analysis (Table 2).

Within the subgroup of individuals with baseline eGFR between 60 to 90 mL/min/1.73 m$^2$, one patient in the TAF group and two patients in the ETV group experienced an eGFR

Peer J

**Table 1 Clinical characteristics of the study population at baseline.**

| Variables | Overall | | | Baseline eGFR≥90 (N = 395) | | | Baseline eGFR<90 (N = 71) | | |
|---|---|---|---|---|---|---|---|---|---|
| | ETV (N = 296) | TAF (N = 170) | P value | ETV (N = 252) | TAF (N = 143) | P value | ETV (N = 44) | TAF (N = 27) | P value |
| Age (years) | 39.5 (31.8–47.8) | 36.5 (32.1–42.7) | 0.019 | 38.0 (31.1–45.8) | 35.5 (31.8–41.7) | 0.074 | 51.3 ± 12.2 | 44.4 ± 10.0 | 0.015 |
| Sex | | | | | | | | | |
| Female | 62 (20.9%) | 55 (32.4%) | 0.009 | 51 (20.2%) | 51 (35.7%) | 0.001 | 11 (25.0%) | 4 (14.8%) | 0.471 |
| Male | 234 (79.1%) | 115 (67.6%) | | 201 (79.8%) | 92 (64.3%) | | 33 (75.0%) | 23 (85.2%) | |
| Hypertension | 23 (7.8%) | 5 (2.9%) | 0.056 | 16 (6.3%) | 3 (2.1%) | 0.098 | 7 (15.9%) | 2 (7.4%) | 0.467 |
| Diabetes mellitus | 24 (8.1%) | 10 (5.9%) | 0.481 | 18 (7.1%) | 8 (5.6%) | 0.700 | 6 (13.6%) | 2 (7.4%) | 0.701 |
| Dyslipidemia | 31 (10.5%) | 15 (8.8%) | 0.679 | 23 (9.1%) | 11 (7.7%) | 0.763 | 8 (18.2%) | 4 (14.8%) | 1.000 |
| MASLD | 70 (23.6%) | 42 (24.7%) | 0.885 | 62 (24.6%) | 37 (25.9%) | 0.873 | 8 (18.2%) | 5 (18.5%) | 1.000 |
| Compensated cirrhosis | 90 (30.4%) | 53 (31.2%) | 0.945 | 72 (28.6%) | 41 (28.7%) | 1.000 | 18 (40.9%) | 12 (44.4%) | 0.964 |
| Antihypertensive agents | 16 (5.4%) | 4 (2.4%) | 0.184 | 10 (4%) | 2 (1.4%) | 0.225 | 6 (13.6%) | 2 (7.4%) | 0.701 |
| Hypoglycemic agents | 19 (6.4%) | 3 (1.8%) | 0.040 | 14 (5.6%) | 3 (2.1%) | 0.171 | 5 (11.4%) | 0 (0%) | 0.149 |
| Hypolipidemic agents | 13 (4.4%) | 3 (1.8%) | 0.217 | 8 (3.2%) | 2 (1.4%) | 0.340 | 5 (11.4%) | 1 (3.7%) | 0.397 |
| Diuretics | 8 (2.7%) | 1 (0.6%) | 0.165 | 6 (2.8%) | 1 (0.7%) | 0.430 | 2 (4.5%) | 0 (0%) | 0.522 |
| Propranolol | 2 (0.7%) | 1 (0.6%) | 1.000 | 2 (0.8%) | 1 (0.7%) | 1.000 | 0 (0%) | 0 (0%) | 1.000 |
| ALT (U/L) | 68.0 (36.0–221.5) | 66.5 (38.0–117.0) | 0.082 | 71.0 (37.8–233.5) | 68.0 (42.5–113.0) | 0.121 | 55.0 (27.5–136.0) | 48.0 (25.5–102.0) | 0.407 |
| AST (U/L) | 45.0 (29.0–113.0) | 41.0 (28.0–70.0) | 0.015 | 46.0 (30.0–117.0) | 43.0 (28.0–67.5) | 0.014 | 38.5 (25.5–69.5) | 35.0 (24.0–67.5) | 0.602 |
| ALB (g/L) | 44.9 (41.0–46.9) | 44.9 (42.9–46.7) | 0.098 | 44.9 (41.1–47.0) | 44.9 (42.9–46.8) | 0.190 | 44.4 (40.2–46.5) | 44.9 (43.1–46.5) | 0.216 |
| TB (μ mol/L) | 15.4 (11.3–23.0) | 14.1 (10.5–18.1) | 0.001 | 15.8 (12.1–23.8) | 14.3 (10.4–17.9) | <0.001 | 12.8 (9.7–20.2) | 13.8 (11.8–21.4) | 0.745 |
| DB (μ mol/L) | 4.8 (3.6–7.7) | 4.3 (3.3–6.0) | 0.002 | 4.9 (3.8–8.4) | 4.3 (3.2–6.0) | <0.001 | 4.1 (3.4–5.9) | 4.6 (3.4–6.5) | 0.836 |
| GGT (U/L) | 41.0 (27.0–95.0) | 34.0 (22.0–55.0) | <0.001 | 45.0 (28.0–98.5) | 33.0 (21.5–55.2) | <0.001 | 38.0 (24.0–77.5) | 40.0 (28.0–53.0) | 0.901 |
| Cr (μ mol/L) | 74.4 ± 13.4 | 74.6 ± 15.1 | 0.881 | 74.0 (64.0–81.0) | 72.0 (60.5–81.0) | 0.501 | 91.0 (76.5–97.5) | 96.4 (91.5–100.0) | 0.101 |

**Table 1** (*continued*)

| Variables | Overall | | | Baseline eGFR≥90 (*N* = 395) | | | Baseline eGFR<90 (*N* = 71) | | |
|---|---|---|---|---|---|---|---|---|---|
| | ETV (*N* = 296) | TAF (*N* = 170) | *P* value | ETV (*N* = 252) | TAF (*N* = 143) | *P* value | ETV (*N* = 44) | TAF (*N* = 27) | *P* value |
| eGFR (ml/min/1.73 m$^2$) | 107.0 (96.6–114.1) | 106.6 (95.3–114.3) | 0.946 | 108.7 (101.2–115.8) | 108.8 (102.0–115.9) | 0.933 | 84.4 (78.3–86.6) | 83.6 (79.6–87.4) | 0.421 |
| HBV DNA | | | | | | | | | |
| Detectable | 268 (90.5%) | 149 (87.6%) | 0.068 | 227 (90.1%) | 127 (88.8%) | 0.153 | 41 (93.2%) | 22 (81.5%) | 0.245 |
| Undetectable | 5 (1.7%) | 0 (0%) | | 5 (2.0%) | 0 (0%) | | 0 (0%) | 0 (0%) | |
| Unknown | 23 (7.8%) | 21 (12.4%) | | 20 (7.9%) | 16 (11.2%) | | 3 (6.8%) | 5 (18.5%) | |
| HBeAg positivity | 144 (48.6%) | 105 (61.8%) | 0.008 | 128 (50.8%) | 92 (64.3%) | 0.012 | 16 (36.4%) | 13 (48.1%) | 0.464 |
| Follow-up duration (weeks) | 86.7 (75.6–98.0) | 86.9 (77.1–94.9) | 0.827 | 86.7 (76.2–98.0) | 86.0 (77.4–92.6) | 0.407 | 86.6 (72.0–99.7) | 94.9 (77.2–98.8) | 0.191 |

**Notes.**

Continuous variables are expressed as mean (standard deviation) or median (interquartile range), according to the distribution of variables. Categorical variables are expressed as number (percentage).

Abbreviations: ALB, Albumin; ALT, alanine aminotransferase; AST, aspartate aminotransferase; Cr, creatinine; DB, direct reacting bilirubin; eGFR, estimated glomerular filtration rate; ETV, entecavir; GGT, gamma-glutamyl transferase; HBeAg, hepatitis B e antigen; HBV, hepatitis B virus; MASLD, metabolic dysfunction-associated steatotic liver disease; TAF, tenofovir alafenamide; TB, total bilirubin.

**Table 2 Factors associated with the development of CKD stage in the subgroup of baseline eGFR ≥90 ml/min/1.73 m².**

| Variables | Univariable COX analyses | | Multivariable COX analyses | |
|---|---|---|---|---|
| | HR (95% CI) | P value | HR (95% CI) | P value |
| Group (TAF *vs.* ETV) | 0.72 (0.39–1.34) | 0.299 | | |
| Age (year) | 1.04 (1.02–1.07) | 0.001 | 1.00 (0.97–1.03) | 0.974 |
| Sex (male) | 3.20 (1.27–8.07) | 0.014 | 2.72 (1.02–7.25) | 0.045 |
| Hypertension | 1.27 (0.40–4.09) | 0.687 | | |
| Diabetes mellitus | 0.94 (0.29–3.02) | 0.915 | | |
| Dyslipidemia | 0.67 (0.21–2.14) | 0.497 | | |
| MASLD | 0.98 (0.51–1.89) | 0.959 | | |
| Compensated cirrhosis | 1.61 (0.91–2.86) | 0.157 | | |
| Antihypertensive agents | 1.33 (0.32–5.47) | 0.694 | | |
| Hypoglycemic agents | 0.96 (0.23–3.96) | 0.958 | | |
| Hypolipidemic agents | 0.36 (0.003–NA) | 0.386 | | |
| Diuretics | 1.24 (0.17–8.97) | 0.833 | | |
| Propranolol | 1.25 (0.01–NA) | 0.880 | | |
| ALT (U/L) | 1.00 (1.00–1.00) | 0.057 | 1.00 (1.00–1.00) | 0.559 |
| AST (U/L) | 1.00 (1.00–1.00) | 0.024 | 1.00 (1.00–1.00) | 0.830 |
| ALB (g/L) | 0.94 (0.89–1.00) | 0.035 | 0.95 (0.89–1.01) | 0.132 |
| DB (μ mol/L) | 1.00 (1.00–1.01) | 0.149 | | |
| GGT (U/L) | 1.00 (1.00–1.00) | 0.194 | | |
| Baseline eGFR (ml/min/1.73 m²) | 0.87 (0.84–0.91) | <0.001 | 0.86 (0.82–0.90) | <0.001 |
| HBV DNA | | | | |
| Undetectable | | | | |
| Detectable | 0.61 (0.08–4.43) | 0.625 | | |
| Unknown | 1.06 (0.13–8.62) | 0.956 | | |
| HBeAg positivity | 0.83 (0.48–1.46) | 0.528 | | |

**Notes.**

Abbreviations: ALB, Albumin; ALT, alanine aminotransferase; AST, aspartate aminotransferase; DB, direct reacting bilirubin; eGFR, estimated glomerular filtration rate; ETV, entecavir; GGT, gamma-glutamyl transferase; HBV, hepatitis B virus; MASLD, metabolic dysfunction-associated steatotic liver disease; TAF, tenofovir alafenamide.

lower than 60 mL/min/1.73 m² during the follow-up, but none had their eGFR decline for more than 12 weeks or to lower than 30 mL/min/1.73 m².

## eGFR dynamic change during follow-up

Within the subgroup of individuals with a baseline eGFR ≥ 90 mL/min/1.73 m², GAMM analysis demonstrated a pattern in which the eGFR initially decreased before stabilizing around week 40 (Fig. 2). The eGFR changes in TAF and ETV groups are presented in Fig. 3. Specifically, the ETV group, exhibited an initial decline in eGFR followed by stabilization, whereas the TAF group experienced a decrease in eGFR initially, followed by a gradual increase after approximately week 30. The eGFR trajectories of the two groups intersected at approximately week 45, indicating differing rates of change between them (Fig. 3). In the initial 0 to 12 weeks, eGFR in the TAF group showed a rapid decline compared to the ETV group (slopes: TAF *vs.* ETV: −0.38 *vs.* −0.15, $p = 0.015$), with a significant difference. From

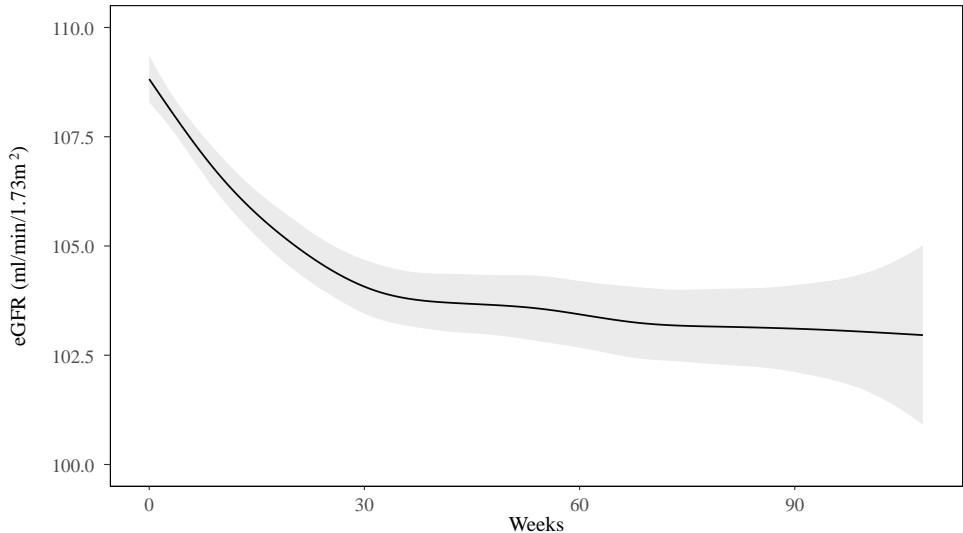

**Figure 2** **Change in eGFR in the subgroup of baseline eGFR ≥ 90 ml/min/1.73 m² in the follow-up.**
GAMM analysis showed a nonlinear change in the eGFR as time went on. The eGFR first decreased and
then around week 40 levelled off.

**Table 3** **Average slope differences of 0 to 12, 12 to 24, 24 to 48, and 48 to 86 weeks between ETV and
TAF groups based on GAMM model.**

| Interval | ETV group | TAF group | P value |
|---|---|---|---|
| 0–12 weeks | −0.15 (−0.23, −0.07) | −0.38 (−0.54, −0.21) | 0.015 |
| 12–24 weeks | −0.14 (−0.20, −0.09) | −0.18 (−0.30, −0.05) | 0.627 |
| 24–48 weeks | −0.10 (−0.14, −0.05) | 0.06 (−0.04, 0.16) | 0.005 |
| 48–86 weeks | −0.03 (−0.07, 0.01) | 0.04 (−0.04, 0.11) | 0.151 |

12 to 24 weeks, the decline in eGFR within the TAF group slowed and became comparable
to that of the ETV group (TAF *vs.* ETV: −0.18 *vs.* −0.14, $p = 0.627$). Between 24 and 48
weeks, the eGFR in the TAF group began to recover, while in the ETV group, the eGFR
continued to decline (TAF *vs.* ETV: 0.06 *vs.* −0.10, $p = 0.005$). Beyond 48 weeks (from 48
to 86 weeks), the eGFR in the TAF group continued to increase slightly, while it stabilized
in the ETV group (TAF *vs.* ETV: 0.04 *vs.* −0.03, $p = 0.151$) (Table 3).

Further analysis revealed that on average, the magnitude of decrease in eGFR from the
baseline was lower in the TAF group ($\beta$: 0.43, 95% CI [0.16–0.69], $p = 0.002$) every 12
weeks. The observed disparity remained statistically significant even after adjusting for
all variables (model 2, $\beta$: 0.40, 95% CI [0.13–0.67], $p = 0.004$) as well as the variables
with $p < 0.05$ (model 3, $\beta$: 0.38, 95% CI [0.11–0.65], $p = 0.006$) (Table 4). Subsequent
sex-stratified analysis revealed significant distinctions between treatment groups among
males (all $p < 0.05$ in three models), while no disparities were evident among females (all
$p > 0.05$ in three models). The age-stratified analysis further demonstrated that individuals
aged 35 to 65 years exhibited significant variances between TAF and ETV groups ($p < 0.05$

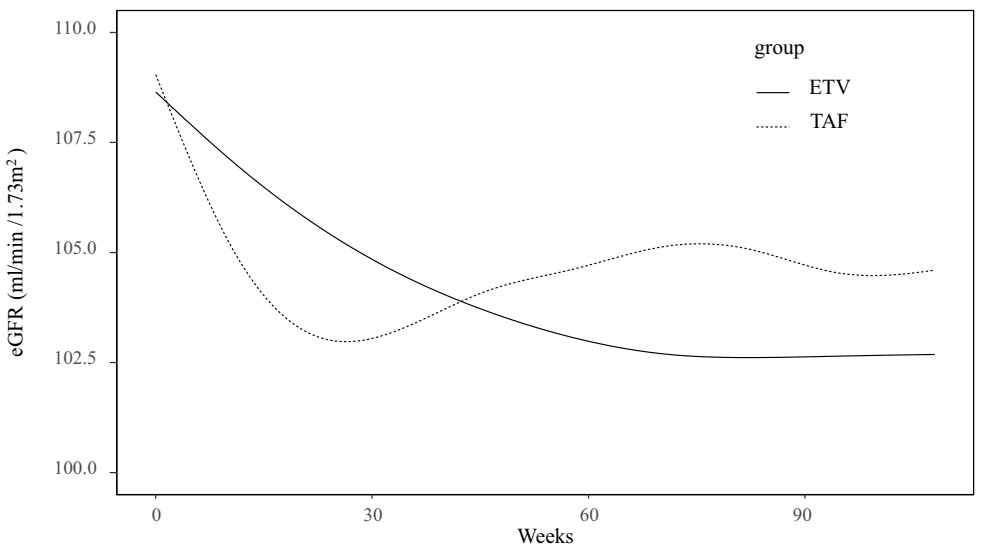

**Figure 3** **Association between eGFR changes and treatment groups in the subgroup of baseline eGFR ≥ 90 ml/min/1.73 m².** Both groups presented nonlinear changes in eGFR.

**Table 4** **The overall difference in eGFR decline from baseline between the two groups per 12 weeks.**

| TAF *vs.* ETV | Model 1 | | Model 2 | | Model 3 | |
|---|---|---|---|---|---|---|
| | β (95% CI) | *P* value | β (95% CI) | *P* value | β (95% CI) | *P* value |
| Baseline eGFR ≥ 90 ml/min/1.73 m² | | | | | | |
| Overall | 0.43 (0.16–0.69) | 0.002 | 0.40 (0.13–0.67) | 0.004 | 0.38 (0.11–0.65) | 0.006 |
| Female | 0.21 (−0.23–0.66) | 0.352 | 0.17 (−0.28–0.63) | 0.451 | 0.19 (−0.26–0.64) | 0.402 |
| Male | 0.37 (0.04–0.70) | 0.026 | 0.35 (0.02–0.068) | 0.039 | 0.33 (0.004–0.66) | 0.047 |
| Age <35 years | 0.26 (−0.17–0.69) | 0.240 | 0.23 (−0.21–0.67) | 0.314 | 0.22 (−0.22–0.65) | 0.330 |
| 35 years ≤ age ≤ 65 years | 0.58 (0.23–0.92) | 0.001 | 0.55 (0.20–0.90) | 0.002 | 0.55 (0.20–0.90) | 0.002 |
| Baseline eGFR <90 ml/min/1.73 m² | | | | | | |
| | 0.34 (−0.31–1.00) | 0.303 | 0.28 (−0.41–0.97) | 0.428 | 0.28 (−0.38–0.95) | 0.406 |

**Notes.**

Model 1: unadjusted model. Model 2: adjusted for all variables. Model 3: adjusted for the variables of which $p < 0.05$ in Model 2. **1.** Baseline eGFR ≥90 ml/min/1.73 m² subgroup: **1.1** overall: adjusted for diuretics, age, sex, baseline eGFR, and HBeAg positivity; **1.2** Female groups: adjusted for age, MASLD, baseline eGFR, DB, and ALB. **1.3** Male group: adjusted for age, diuretics, baseline eGFR, and HBeAg positivity; **1.4** age <35 years groups: adjusted for baseline eGFR. **1.5** 35 years ≤ age ≤ 65 years group: adjusted for diuretics, baseline eGFR, and DB. **2.** Baseline eGFR < 90 ml/min/1.73 m² subgroup: adjusted for cirrhosis and baseline eGFR.

in three models), whereas those under 35 years did not display any differences (all $p > 0.05$ in three models).

The interaction between treatment groups and sex (male *vs.* female) was not statistically significant (all $p > 0.05$ in three models) (Table S1). The interaction between treatment groups and age (≥ 35 *vs.* <35 years) demonstrated a significant result in the unadjusted model ($\beta$: 3.81, 95% CI [0.10–7.53], $p = 0.045$). However, this effect showed no significance after adjustment ($p > 0.05$ in adjusted models) (Table S1).

In the subgroup of baseline eGFR between 60 to 90 mL/min/1.73 m$^2$, GAMM analysis demonstrated a linear change in eGFR (Fig. S2). The rate of eGFR decline varied between the two treatment groups (Fig. S3), but it did not reach a statistically significant difference ($\beta$: 0.34, 95% CI [$-0.31$–1.00], $p = 0.303$) (Table 4).

### Sensitivity analyses

In the sensitive analyses, the piecewise LMM showed similar findings as the GAMM (Table S2). Among the individuals with complete baseline data, the multivariable Cox regression indicated similar results that sex and baseline eGFR were independently associated with CKD development in the subgroup of baseline eGFR $\geq$ 90 mL/min/1.73 m$^2$ (Table S3).

GAMM analyses also demonstrated consistent patterns and overall differences of eGFR changes between TAF and ETV groups in the following scenarios: (1) individuals with complete baseline data (Figs. S4–S7, Table S4); (2) individuals who had no missing eGFR measurements in the first 12 (Figs. S8–S9, Table S5) and 30 weeks (Figs. S10–S11, Table S6); (3) individuals with at least three eGFR assessments over the periods of 0 to12, 12 to 24 and 24 to 48 weeks (Figs. S12–13, Table S7). However, among individuals without missing eGFR in the first 30 weeks, no significant difference was observed in the male subgroup.

## DISCUSSION

This study assessed the progression of CKD stage and the dynamic changes of follow-up eGFR in a retrospective cohort, comparing TAF and ETV as antiviral therapies for treatment-naïve individuals with CHB. In terms of kidney outcomes, there was no difference in the progression of CKD stage during follow-up between TAF and ETV (9.8% vs. 13.9%) in the subgroup of baseline eGFR $\geq$ 90 mL/min/1.73 m$^2$. However, there were notable differences in eGFR dynamic changes between TAF and ETV during follow-up, particularly among individuals with baseline eGFR $\geq$ 90 mL/min/1.73 m$^2$. The TAF group showed a lesser decline in eGFR over 12 weeks compared to the ETV group.

Within the ETV group, the incidence of CKD development was 13.9%, aligning closely with existing literature reports ranging from 9.1% to 19.7% (Lampertico et al., 2020; Lin et al., 2017; Wong et al., 2018). The rates of CKD stage progression varied across studies, possibly attributed to variances in baseline characteristics among enrolled patients. Wong et al. reported a 19.7% rate of CKD progression among individuals with a higher average age of 51 years (Wong et al., 2018), whereas the median age in our study was 38.0 years, potentially leading to different results.

In this study, the progression of the CKD stage had no difference between TAF and ETV groups (9.8% vs. 13.9%, $p = 0.304$) among the individuals of a baseline eGFR >90 mL/min/1.73 m$^2$. Over the last decade, some studies have investigated the kidney safety advantage of TAF over ETV. However, a definitive conclusion has yet to be reached. A retrospective study from South Korea demonstrated no difference in eGFR change at 48 weeks between TAF and ETV (1.90 vs. $-0.40$ mL/min/1.73 m$^2$, $p = 0.055$) (Jeong, Shin & Kim, 2021). Another retrospective study in South Korea demonstrated that ETV was linked to an increased risk of CKD development compared to TAF (Jung et al., 2022a). Since acute-on-chronic liver failure (ACLF) is likely to develop hepatorenal syndrome, several

studies have also been carried out in CHB patients with ACLF to compare kidney safety between the two treatments. *Peng et al. (2023)* conducted a study examining kidney safety in CHB-related ACLF treated with TAF and ETV for four weeks and found that TAF had a lower risk of kidney function reduction. Another prospective study involving CHB-related ACLF reported no difference in eGFR changes between the two drugs (*Li et al., 2021*). A meta-analysis (*Liu et al., 2023*), including studies on treatment-naïve individuals and those switching from TDF, found no difference in eGFR decrease between TAF and ETV. Nevertheless, the serum creatinine elevated more in the TAF group as opposed to the ETV group. The authors suggested further studies to explore kidney safety during treatment with TAF and ETV.

In this study, the male sex (adjusted HR 2.72, 95% CI [1.02–7.25], $p = 0.045$) and baseline eGFR (adjusted HR 0.86, 95% CI [0.82–0.90], $p < 0.001$) were identified as independent risk factors for the CKD development. These findings align with previous literature. A retrospective cohort study from Hong Kong, China, reported a higher risk of kidney impairment and progression in men compared to women (HR: 2.67, 95% 2.16~3.29, $p < 0.001$) (*Mak et al., 2022*). It is widely recognized that baseline kidney function is an independent risk factor for CKD progression during oral antiviral agents (*Qi et al., 2015*; *Shin et al., 2016*; *Trinh et al., 2019*). These results suggest that from the perspective of kidney safety, sex, and baseline eGFR should be considered before initiating antiviral therapy.

Additionally, monitoring temporal eGFR changes during treatment should not be neglected in practice. Our study also identified a non-linear decrease in eGFR among the individuals with baseline eGFR $\geq$ 90 mL/min/1.73 m$^2$. The eGFR significantly declined from the baseline within the initial 30 weeks, followed by a trend toward stabilization (Fig. 2). Further GAMM group analysis indicated distinct patterns in eGFR changes between TAF and ETV. While the eGFR initially decreased and gradually increased in the TAF group, it decreased and tended to stabilize in the ETV group. The TAF group experienced a rapid decline of eGFR during the first 12 weeks, followed by a slower decline in 12 to 24 weeks and then a recovery starting from 24 to 48 weeks. This pattern supports the idea that the reduction in eGFR with TAF is temporary, with improvements beginning around week 30, as illustrated in Fig. 3. In contrast, the ETV group showed a more gradual but persistent decline in eGFR until it stabilized around week 40. The piecewise LMM analysis revealed a transient eGFR increment in the TAF group from weeks 24 to 48 with a slope of 0.09 mL/min/1.73 m$^2$. In contrast, the ETV group experienced a continued decline in eGFR, by a slope of $-0.10$ mL/min/1.73 m$^2$ ($p < 0.001$) (Table S2). This improvement in the TAF group may reflect a recovery from the initial tubular toxicity, possibly due to TAF's lower systemic exposure stabilizing tenofovir levels by this period (*Ray, Fordyce & Hitchcock, 2016*).

The eGFR change pattern observed in the ETV group aligns with Tsai's study, which reported that in patients with baseline eGFR $\geq$ 90 mL/min/1.73 m$^2$, eGFR declined most significantly during the first year of ETV treatment and then stabilized in the second year (*Tsai et al., 2016*). ETV is generally regarded as safe for the kidney; however, its renal effects may involve mild direct toxicity to proximal tubular cells or accumulation in patients

with pre-existing renal impairment. A case report highlighted a 73-year-old patient who developed Fanconi syndrome after five years of ETV use, with renal function improving after switching to TAF (*Fujii et al., 2019*).

In our study, we noted a different eGFR trajectory between TAF and ETV. TAF is a prodrug of tenofovir, designed to minimize systemic toxicity. However, TAF can still cause an initial decline in eGFR, as observed in our findings. This decline is likely due to mild mitochondrial toxicity affecting the proximal tubular cells. Tenofovir is actively transported into these cells *via* organic anion transporters (OAT1 and OAT3), where it can accumulate and impair mitochondrial function, leading to tubular dysfunction and a decrease in eGFR (*Ray, Fordyce & Hitchcock, 2016*). This is further supported by a case report involving TAF-related nephrotoxicity in an HIV patient, wherein a kidney biopsy revealed acute tubular injury and mitochondrial damage after three months of TAF treatment (*Ueaphongsukkit et al., 2021*). The transient eGFR decline in the TAF group, followed by an improvement, aligns with its 90% lower systemic exposure than TDF. This lower exposure allows tubular cells to recover through adaptive mechanisms or reduced toxicity (*Ray, Fordyce & Hitchcock, 2016*). These findings emphasize the importance of renal monitoring, particularly in TAF- and ETV-treated patients who have renal risk factors.

Early detection of proximal tubular dysfunction, which is potentially the underlying cause of declines in eGFR, can be achieved using markers such as urine phosphorus, fractional excretion of phosphate ($FEPO_4$), the Bijvoet equation (TmP/GFR), and fractional excretion rate of uric acid (FEUA). Elevated $FEPO_4$ and FEUA, or a reduced TmP/GFR, often occur before eGFR decline in patients treated with tenofovir (*Lee et al., 2020*; *Brayette et al., 2020*). These markers could theoretically indicate an early rapid decline in the TAF group, though they were not evaluated in this study.

Although the CKD development had no difference between the two agents, GAMM analysis demonstrated overall varying rates of changes in eGFR every 12 weeks between the two groups. This difference remained significant when adjusting for age, sex, diuretic usage, baseline eGFR, and HBeAg positivity. These findings suggest that the eGFR decline associated with TAF might be transient. Additionally, given that male sex was identified as one of the risk factors of CKD stage progression in our study, a sex-stratified analysis was also conducted. While the difference was not as significant as that in the overall group, the eGFR decrease in the TAF group remained smaller compared to that in the ETV group ($p = 0.047$) among males. However, no noted difference was observed among females. Furthermore, an age-stratified analysis revealed that among individuals aged 35 to 65, the magnitude of the decline in eGFR per 12 weeks in the TAF group was notably less than that in the ETV group (adjusted $\beta$: 0.55, 95% CI [0.20–0.90], $p = 0.002$). Despite the age stratification in our study being 35 years, approximately the median age at baseline, our study findings align with the recommendation from the EASL guideline advocating for safer medications for patients aged older than 60 (*European Association for the Study of the Liver, 2017*). Besides, aging causes progressive reductions in eGFR and kidney blood flow (*Weinstein & Anderson, 2010*), rendering eGFR more susceptible to drug toxicity in older individuals. Previous research has demonstrated that eGFR can be preserved

up to the age of 40 years, marking the point at which age-dependent kidney loss begins (*Pottel et al., 2016*). Individuals typically have an average eGFR of 107.3 mL/min/1.73 m$^2$ before 40 years old, followed by a yearly decrease rate of 0.91 mL/min/1.73 m$^2$ (*Pottel et al., 2017*).

To examine potential effect modification, we conducted interaction analyses between treatment groups and both sex and age as a secondary analysis. The results revealed non-significant interactions after adjustments, indicating that the treatment effect does not vary significantly by sex or age. However, significant subgroup outcomes observed in males and individuals aged 35 to 65 years support the biological understanding that kidney function tends to decline more rapidly in men than in women (*Carrero et al., 2018*), and that aging is associated with a decrease in renal function. This suggests a potential clinical advantage of TAF for these specific subgroups. The discrepancy between the interaction and subgroup results could be attributed to confounding factors. Nonetheless, the consistency across models of the stratified findings indicates that the usage of TAF over ETV in individuals over 35 with normal kidney function, especially among males, contributes to preserving kidney function against aging-related decline.

Our study also included individuals with an eGFR ranging from 60 to 90 mL/min/1.73 m$^2$ at baseline. A small number of cases underwent a reduction in eGFR to below 60 mL/min/1.73 m$^2$, but none of these patients in either group experienced a decline to below 30 mL/min/1.73 m$^2$. These reductions were temporary, lasting less than 12 weeks. Unlike the subgroup of baseline eGFR ≥ 90 mL/min/1.73 m$^2$, the changes in eGFR over time were linear. Although differences were shown in the extent of eGFR alteration between the two agents, no statistically significant difference was detected. These findings suggest that these variations may fall within the range of natural variation, and both TAF and ETV exhibit a satisfactory safety profile for individuals with baseline eGFR levels ranging from 60 and 90 mL/min/1.73 m$^2$.

This study had some limitations. Firstly, it could not rule out bias in drug selection potentially influenced by patients' kidney risk profiles as the retrospective designed. However, we employed multivariate Cox regression and GAMM models to adjust for the major confounding factors. Secondly, because of the retrospective study nature, serum creatinine was the only biomarker of kidney function that could be fully obtained. Including data on early tubulopathy markers, such as urine phosphorus, FEPO4, TmP/GFR, and FEUA would help clarify the mechanisms behind the eGFR trajectories when treating with TAF or ETV. Thirdly, given that TAF has recently been introduced in China, our follow-up period was limited to approximately two years, whereas CHB treatment typically lasts for decades to a lifetime. Lastly, although patients with baseline eGFR lower than 90 mL/min/1.73 m$^2$ were included, the sample size was restricted.

## CONCLUSION

To summarize, our study findings suggest that the kidney safety profile of TAF in treatment-naïve individuals with CHB is comparable to ETV, without significant differences in CKD progression. The stratified analyses reveal the potential benefits of TAF over ETV,

particularly for male patients or those aged 35 and above. Additionally, the eGFR trajectories highlight the importance of early renal monitoring during CHB treatment. Future research should consider adopting a prospective or randomized design and extending follow-up periods to gain more comprehensive insights into the kidney impact of TAF in CHB individuals.

### Funding

This work was supported by hospital projects of Shenzhen Third People's Hospital Research Fund (No: 24250G1006, G2022110, and 22244G4001), and Shenzhen Science and Technology Research and Development Fund (No: JCYJ20190809143609762). The funders had no role in study design, data collection and analysis, decision to publish, or preparation of the manuscript.

### Grant Disclosures

The following grant information was disclosed by the authors:
Shenzhen Third People's Hospital Research Fund: 24250G1006, G2022110, 22244G4001.
Shenzhen Science and Technology Research and Development Fund: JCYJ20190809143609762.

### Competing Interests

The authors declare there are no competing interests.

### Author Contributions

- Xuan Li conceived and designed the experiments, authored or reviewed drafts of the article, and approved the final draft.
- Qiang Wu analyzed the data, prepared figures and/or tables, and approved the final draft.
- Fang Huang performed the experiments, authored or reviewed drafts of the article, and approved the final draft.
- Changxiang Lai performed the experiments, authored or reviewed drafts of the article, and approved the final draft.
- Fengjuan Chen performed the experiments, authored or reviewed drafts of the article, and approved the final draft.
- Juan Meng analyzed the data, prepared figures and/or tables, and approved the final draft.
- Fang Wang analyzed the data, prepared figures and/or tables, and approved the final draft.
- Hui Zeng conceived and designed the experiments, authored or reviewed drafts of the article, and approved the final draft.
- Lina Zhang conceived and designed the experiments, authored or reviewed drafts of the article, and approved the final draft.

## Human Ethics

The following information was supplied relating to ethical approvals (*i.e.*, approving body and any reference numbers):

The Ethics Committee of Shenzhen Third People's Hospital granted Ethical approval to carry out the study within its facilities (Ethical Ref: 2022-142-02, 2022-142-03).

## Data Availability

Raw data is available in the Supplemental Files.

## Supplemental Information

Supplemental information for this article can be found online at http://dx.doi.org/10.7717/peerj.19901#supplemental-information.

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
