# Peer review of "Longitudinal changes in estimated glomerular filtration rates in chronic hepatitis B patients treated with Tenofovir Alafenamide *vs* Entecavir"

_PeerJ, doi:10.7717/peerj.19901_

## Round 0.1 · original submission · Major Revisions

The study methodology should be explain in more detail. In addition, more in depth comments are needed in the discussion section. Also, please pay attention to the study limitations.

·

Basic reporting

1-1. Please consider using the terminology "kidney" rather than "renal" across the whole manuscript (i.e., renal safety → kidney safety).
1-2. [Methods, Line 89~96] I suggest the authors clearly provide detailed definitions of the exclusion criteria. Did the authors define "presence of chronic kidney disease" as the baseline eGFR under 60? What is the definition of "severe diseases of other systems?" Does "missing data on follow-up eGFR" mean no follow-up eGFR measurements?

Experimental design

2-1. [Methods, Line 94~95] I wonder why the authors excluded individuals with death during follow-up, rather than analyzing eGFR as a death-censored outcome. Kindly explain the rationale of this exclusion criteria and qualitatively review the possibility of selection bias.
2-2. [Methods, Line 128~129] I would like to ask why the authors conducted single imputation to median values rather than multiple imputation, despite the MAR assumption. Furthermore, MAR is rarely satisfied under the real-world circumstances. Therefore, I recommend presenting a sensitivity analysis result from "complete case analysis" under the MNAR with a conditional independence assumption.

Validity of the findings

3-1. [Results] Is there any differentiality of missing in eGFR measurements around the time point of 12 weeks? As differential missingness could lead to the initial rapid eGFR decline observed only in the TAF group, I recommend the authors address the non-differential missing rate of eGFR over the period of 0~30 weeks.
3-2. [Methods, Line 88] Why did the authors utilize a liberal inclusion criterion on the number of creatinine measurements? A single follow-up creatinine measurement per individual could lead to an unstable estimation in eGFR slope, as it typically requires at least three follow-up measurements. If possible, I recommend the authors present a sensitivity analysis result by restricting individuals with at least three measurements over the periods of 0~12w, 12~24w, and 24~48w and assess whether the results were replicated.
3-3. [Methods, Results, Discussion] Please specify more detailed methods for comparing the rate of eGFR decline on a 12-week scale. While the authors only explained GAMM analysis with a smoothing curve, it seemed to me that the authors might have assumed "a linear" time effect on eGFR to derive "a single" coefficient contrast across two treatment groups. As per Figure 3, while the ETV group exhibited a persistent and slower decline over the period of 0~60 weeks, the TAF group showed a more rapid decline across 0~12 weeks followed by comparable eGFR values thereafter. Since the linear mixed effects modeling focuses on estimating average expected values, I believe the results on Line 199 might be influenced by the longer time of initial eGFR decline, even though the long-term eGFR levels are similar. Therefore, to robustly claim that TAF is preferable to ETV from the perspective of kidney prognosis (as described in Discussion Line 279~281), I suggest the authors provide RM-ANOVA results or LMM with splines for the time variable.

Additional comments

4-1. [Discussion] I suggest that the authors review any possible pharmacological rationale that could explain an initial decline in eGFR from using TAF or ETV to support the biological plausibility of their findings.

Reviewer 2 ·

Basic reporting

I read with attention Li et al's manuscript concerning kidney-injury-associated to HBV treatment.
The text is well-written with a sufficient English level.
Some comments can be made:
- Be careful in introduction section: it si superflu if you report indications of treatment.
- In M&M, replace creatine by creatinine.
- In discussion and moreover in references, specific informations above early detection methods for tubulopathy markers (such as urine phosphore, Bijvoet equation, etc..) could be more interesting than three words in limitis section.

Experimental design

Cf. previous.
I recognize I don't know very well GAMM.

Validity of the findings

Cf. before.

Additional comments

None.

---

## Round 0.2 · Minor Revisions

Some expressions must be changes for a better understanding, and more important, additional details are needed for the statistical analysis and clarifications on some results (please see the reviewer's comments).

·

Basic reporting

#1. Some expressions should be edited regarding the reporting standards:
- Any dates should be expressed in "ordinal numbers" (i.e., Line 87: August 31, 2022 → August 31st, 2022.)
- Line 91, 157, ...: Range should be expressed using an en dash (–,) not a hyphen (-.)
- Line 95, 117: ">=" should be edited by "≥."
- Line 98: "mL" is omitted in the unit of baseline eGFR.
- Line 120, ...: Nonalcoholic Fatty Liver Disease (NAFLD) should be now edited as Metabolic Dysfunction-Associated Steatotic Liver Disease (MASLD.)
- Line 129, 168, ...: Numbers and units should always be written with spaces (i.e., 35g/L → 35 g/L, ≥90mL/min/1.73m2 → ≥90 mL/min/1.73 m2.)

Experimental design

#2. I think the authors could elaborate more on their sex- and age-stratified analyses by including an interaction term between treatment allocation (ETV vs TAF) and potential effect modifier (male vs female.) While the authors qualitatively suggested sex and age as effect modifiers through simply comparing statistical significance of coefficients by stratification, additional interaction analyses could "quantitatively" reveal the significance of effect modification as a multiplicative scale.

Validity of the findings

#3. Please provide any possible explanations regarding the results from piecewise LMM on week 24-48. These results could indicate a transient increment in eGFR in the TAF group during week 24-48, and the TAF group's eGFR slope during week 24-48 was statistically different from the ETV group's.

Additional comments

I thank the authors for their kind response. Most of my concerns at round 1 are adequately addressed in the revised manuscript, and I only have a few minor points requesting authors to modify.

---

## Round 0.3 · accepted · Accept

The minor suggestions of the reviewer in the second round of revision were adequately resolved by the authors. Therefore, I consider the current version of the manuscript suitable for publication.